TOPICAL REVIEW

# Short-term exercise-induced protection of cardiovascular function and health: why and how fast does the heart benefit from exercise?

Dick H. J. Thijssen[1,2] 🆔, Laween Uthman[1,3] 🆔, Yasina Somani[2] and Niels van Royen[3]

[1]*Radboud Institute for Health Sciences, Departments of Physiology, Nijmegen, The Netherlands*
[2]*Research Institute for Sport and Exercise Sciences, Liverpool John Moores University, Leicester, UK*
[3]*Cardiology, Radboud University Medical Center, Nijmegen, The Netherlands*

Edited by: Ian Forsythe & Simon Gandevia

The peer review history is available in the Supporting Information section of this article (https://doi.org/10.1113/JP282000#support-information-section).

**Abstract** Regular exercise training has potent and powerful protective effects against the development of cardiovascular disease. These cardioprotective effects of regular exercise training are partly explained through the effects of exercise on traditional cardiovascular risk factors and improvement in cardiac and vascular health, which take several weeks to months to develop. This review focuses on the observation that single bouts of exercise may also possess an under-recognized, clinically useful form of immediate cardioprotection. Studies, performed in both animals and humans, demonstrate that single or short-term exercise-induced protection (SEP) attenuates the magnitude of cardiac and/or vascular damage in response to prolonged ischaemia and reperfusion injury. This review highlights preclinical evidence supporting the hypothesis that SEP activates multiple pathways to confer immediate protection against ischaemic events, reduce the severity of potentially lethal ischaemic myocardial injury, and therefore act as a physiological first line of defence against injury. Given the fact that the extent of SEP could be modulated by exercise-related and

**Dick H. J. Thijssen's** work is focused on the prevention of cardiovascular disease. His work focuses on exploring and understanding mechanisms explaining the benefits of exercise training. He examines the mechanisms and potential clinical relevance of short-term exercise to mediate clinically relevant benefits. Moreover, he explores the benefits of long-term exercise training, which are linked to improvement in micro- and macrovessel function and structure, and factors that moderate these effects (e.g. age, sex). His work also aims to understand haemodynamic stimuli (e.g. shear stress) that mediate improved vascular function and structure. Ultimately, this contributes to (non-)pharmacological strategies in the management of patients with cardiovascular risk and/or disease.

subject-related factors, it is important to recognize and consider these factors to optimize future clinical implications of SEP. This review also summarizes potential effector signalling pathways (i.e. communication between exercising muscles to vascular/cardiac tissue) and intracellular pathways (i.e. reducing tissue damage) that ultimately confer protection against cardiac and vascular injury. Finally, we discuss potential future directions for designing adequate human and animal studies that will support developing effective SEP strategies for the (multi-)diseased and aged individual.

(Received 18 June 2021; accepted after revision 10 December 2021; first published online 21 December 2021)

**Corresponding author** Dick H. J. Thijssen: Research Institute of Sport and Exercise Sciences, Liverpool John Moores University, Tom Reilly Building, L3 3AF, Liverpool, UK. Email: d.thijssen@ljmu.ac.uk

**Abstract figure legend** Single to short-term exercise preconditioning (SEP) stimulates the release of circulating factors and activation of intramyocardial signalling, resulting in attenuated severity of myocardial ischaemia reperfusion (IR) injury. As opposed to SEP, regular exercise (weeks, months, years) also prevents IR injury episodes by inhibiting the development and/or progression of atherosclerosis. This process is mediated by cardiovascular structural and functional changes and reductions in cardiovascular risk factors following regular exercise.

### Key points

- Single or short-term exercise-induced protection (SEP) attenuates the magnitude of cardiac and/or vascular damage in response to prolonged ischaemia and reperfusion injury (IR injury).
- SEP activates multiple pathways to confer cardiac protection, which develops remotely at the site of the activated muscle by release of circulating molecules, which transfer towards activation of intramyocardial signalling that promotes cell survival during episodes of IR injury.
- SEP represents an attractive intervention in aged individuals and in those with co-morbidities. The immediate protection, low cost and simplicity to increase the 'dose' of SEP offers unique opportunities in the clinical applications of SEP.

## Introduction

Cardiovascular disease (CVD) continues to be the leading cause of morbidity and mortality worldwide (World Health Organization, 2021). At the same time, half of the adult population does not meet the international criteria for sufficient physical activity (Guthold *et al.* 2018; Piercy, 2019). Moreover, it is expected that across the next few decades the burden of CVD and the physical inactivity pandemic are likely to worsen (McClellan *et al.* 2019). These key challenges that we currently face may be inextricably linked, especially when considering the relatively high prevalence of physical inactivity in subjects with CVD compared to the general population. Knowledge of the benefits of physical activity dates back to the sound advice of Hippocrates, who suggested that some exercise, but not in excess, was 'the safest way to health'. Our empirical understanding that confirmed these benefits of physical activity in the prevention of CVD, however, only began in the early 1950s with studies from Professor Morris' laboratory. His research team compared jobs with different physical demands, and found higher prevalence of CVD in jobs linked to lower physical activity demands (Morris *et al.* 1953, 1966). Currently, robust evidence supports the impact of regular physical activity and exercise in primary and secondary prevention of CVD (Shiroma & Lee, 2010).

Understanding the benefits of exercise training in the prevention of cardiovascular mortality and morbidity is important to further optimize the prescription of 'exercise as medicine'. The reduction in the occurrence of cardiovascular events with exercise can only partly be attributed to the direct effects of exercise on traditional cardiovascular risk factors, such as hypertension, hyperlipidaemia and insulin resistance (Mora *et al.* 2007; Green *et al.* 2017). Although exercise training improves individual risk factors, when taking these benefits together ~27–41% of the benefits of regular physical activity can be explained by risk reduction (Taylor *et al.* 2006; Mora *et al.* 2007; Hamer *et al.* 2012). Increasing evidence supports a role for favourable adaptations in cardiac and vascular structure and function to explain this 'risk factor gap' (Green *et al.* 2008, 2017; Thijssen *et al.* 2016). However, these changes in risk factors and cardiovascular function or structure may take several weeks or up to months to establish, suggesting a similarly long period before cardioprotection through regular exercise training is present. However, increasing evidence suggests that a single or few bouts of exercise already offer cardiovascular protection,

which we refer to as single or short-term, exercise-induced protection (SEP). The benefits of SEP may relate to reducing the magnitude of injury that occurs during (ischaemic) events (Fig. 1 ). This suggests that exercise training may have immediate, short-term beneficial effects through SEP that attenuate injury induced by cardio-vascular insults (Thijssen *et al.* 2018).

This review endeavours to summarize the evidence and proposed mechanisms related to SEP; the concept that single or repeated episodes of exercise prior to prolonged ischaemia and subsequent reperfusion can induce cardio-protection (Thijssen *et al.* 2018). To this end, we first summarize the evidence supporting the concept of SEP, derived from both direct and indirect examples in animal and human models of myocardial injury. Second, we focus on the effects of factors such as exercise intensity and mode, prior training status, and age. Third, we summarize potential underlying mechanisms of SEP, related to both the signal transduction pathways and cellular pathways of protection against apoptosis. Finally, we provide a discussion on future directions, specifically related to strategies and scientific areas where SEP can improve outcomes, and how to optimize recommendations for implementation in clinical populations.

## Preconditioning: immediate protection against ischaemia–reperfusion injury through exercise

Myocardial ischaemia–reperfusion (IR) injury represents a major contributor to cardiovascular-related morbidity and mortality (Hausenloy *et al.* 2016; Heusch, 2020). While reperfusion is essential for restoring blood flow to cardiac tissue and preventing tissue necrosis from ischaemia, e.g. during a myocardial infarction or cardiac surgery, the rapid reintroduction of blood flow paradoxically causes greater myocardial injury (Powers *et al.* 2007). The concept of ischaemic preconditioning (IPC) by inducing brief periods of ischaemia prior to prolonged myocardial ischaemia was introduced in 1986 by Murry's lab (Murry *et al.* 1986). A first indication towards remote IPC was apparent when brief episodes of ischaemia in the circumflex branches of the coronary vasculature resulted in a smaller infarct size following

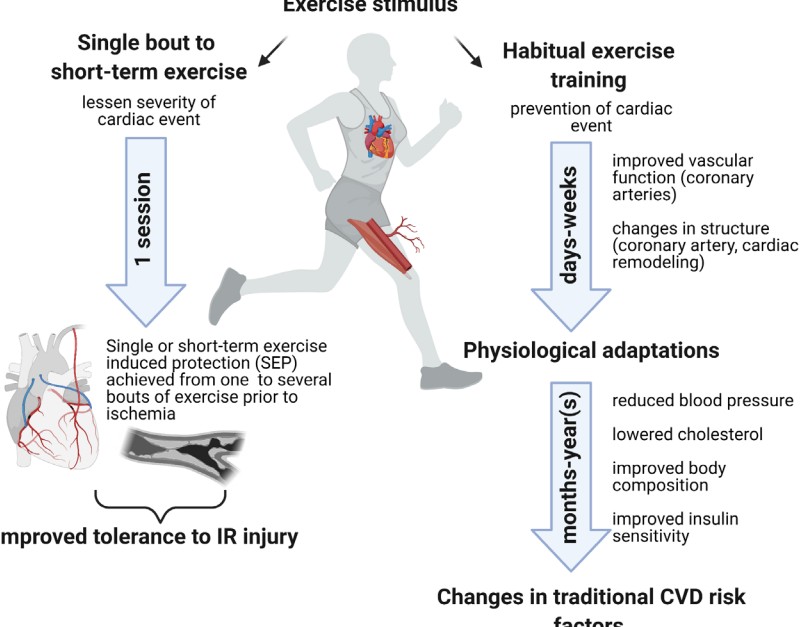

**Figure 1. Overview of benefits of short-term and habitual exercise in the prevention of cardiovascular disease**

Single or short-term exercise-induced protection (SEP) against ischaemia–reperfusion (IR) injury may lessen the severity of myocardial injury from cardiac surgery or myocardial infarction, while habitual exercise training (days–years) leads to physiological adaptions and changes in traditional cardiovascular disease (CVD) risk factors that may prevent the occurrence of a cardiac event.

  

sustained occlusion of the left anterior descending artery (Przyklenk *et al.* 1993). This stimulated investigations into the clinical application of IPC by remote episodes of ischaemia to protect the target organ during prolonged ischaemic events. The traditional stimulus of short, repeated (remote) ischaemia shows some remarkably similarity with exercise. Yamashita *et al.* (1999) first demonstrated that a single bout of treadmill exercise in rats leads to a 60% reduction in infarct size. A limited number of studies have confirmed these effects of SEP preceding IR injury in rats and dogs (Yamashita *et al.* 1999; Domenech *et al.* 2002; Hoshida *et al.* 2002; Parra *et al.* 2010, 2015), as summarized in Table 1. Interestingly, this protection was maintained for up to 9 days following cessation of a 3 day training period (Lennon *et al.* 2004*a*). Likewise, cardioprotective effects have been observed following short-term daily exercise repeated for up to 7 days (Akita *et al.* 2007; McGinnis *et al.* 2015). Aiming to better understand these cardioprotective effects of exercise, some preclinical studies have explored exercise characteristics. Lennon *et al.* (2004*b*) compared three consecutive days of interval exercise in rats, either at moderate or high intensity, and reported that myocardial damage following IR was equally reduced by either modality. Taken together, these preclinical studies clearly show the ability of short-term exercise to induce immediate protection.

Cardioprotection through IPC shows a remarkable, biphasic pattern of protection; the first phase manifests almost immediately following IPC and persists over several hours while the late phase appears after 12–24 h and may last up to 72 h (Kuzuya *et al.* 1993; Hausenloy & Yellon, 2010). Interestingly, SEP seems to protect (cardio)vascular tissue in a similar pattern. The course of protection following a single treadmill exercise manifests in a biphasic manner, consisting of an early phase first window (minutes to hours) offering strong protection and a delayed second window of mild protection (days) (Yamashita *et al.* 1999; Domenech *et al.* 2002). The same authors subsequently reported that protection against IR injury reappeared 24 h following an acute bout of exercise in rats (Yamashita *et al.* 2001). Other animal studies have provided further proof of these cardioprotective effects and the biphasic nature of a single bout of exercise (Domenech *et al.* 2002; Michelsen *et al.* 2012; Parra *et al.* 2015). This suggests that SEP and IPC share some similarities in the magnitude and pattern of protection, which may aid in exploring underlying pathways or to enhance preconditioning effects.

Preclinical studies have further explored potential differences between single and repeated bouts of exercise on the magnitude of cardiac protection. For example, conducting two exercise bouts with a 48 h interval exhibited cardioprotection in a continuous monophasic manner, persisting from 30 min until 60 h following

exercise in rats (Hoshida *et al.* 2002). These results suggest that greater cardioprotection is obtained conducting two *vs.* one exercise bout (∼84% *vs.* ∼58% reduction in infarct size; estimated from data representation in the figures), although more research would be required to validate this observation. Similar monophasic protection by short-term repeated exercise has been observed against neurovascular injury and brain infarct size following stroke in rats undergoing high-intensity interval training (HIIT) (Hafez *et al.* 2020). Another preclinical study performed direct comparison of 3 days *vs.* 3 weeks of intermittent running in rats and also found comparable protection from isoproterenol-induced myocardial injury (Sun & Pan, 2014). These observations suggest that the protective effects of exercise, i.e. the attenuated tissue damage in response to injury, is immediately present upon single exercise and remains present to a similar extent when continuing exercise. An important difference between single and short-term exercise is that the 'nadir', i.e. the drop in protection after 2–3 h following single exercise, disappears upon subsequent exercise bouts. Consequently, the effects of SEP may be prolonged with repeated exercise bouts, thereby offering continuous protection against IR injury. Collectively, these animal-based studies have demonstrated the ability of short-term exercise to induce (remote) protection of cardiac tissue against IR injury. These benefits of SEP seem time-dependent, whereas the effect is robust as it remains present upon repeated bouts of exercise.

## Does SEP also prevent cardiac and vascular injury in humans?

Where preclinical animal models of cardiac injury are typically achieved through binding a selected coronary artery or making the entire heart ischaemic, such models are unavailable in humans. However, clinical observations in humans may offer some insight. For example, previous observations in patients with angina revealed that a single session of moderate- to high-intensity exercise caused an attenuation in exercise-induced ischaemia upon the second exercise effort (Williams *et al.* 2014). This phenomenon, termed 'warm-up angina', is supported by studies reporting a delay in the onset as well as attenuation of the ST segment depression during a sequential exercise bout and this protection appears to act in a biphasic manner, sharing features of SEP (Paraskevaidis *et al.* 2005; Williams *et al.* 2014; Lalonde *et al.* 2015). Other clinical observations revealed that higher physical activity level in the week prior to a myocardial infarction or coronary bypass surgery was related to lower cardiac-related mortality following the cardiac event (Abete *et al.* 2001; Rengo *et al.* 2007, 2010). Finally, a clinical study evaluated the effect of three bouts of aerobic interval exercise

**Table 1. Summary of the animal studies demonstrating the efficacy of SEP in different endurance exercise and IR protocols**

| | General information | | | | | Exercise protocol | | | | | Injury model | | Outcomes | |
|---|---|---|---|---|---|---|---|---|---|---|---|---|---|---|
| Reference | Year | Species | Sex[1] | Strain | Repeat | Period[2] | Speed (m/min)[3] | Intensity (%V̇O2max) | Time since exercise | Ischaemia | Reperfusion | Smaller infarct size vs. control | Mechanism |
| Hoshida | 2002 | Rat | M | Wistar | 1 vs. 2 | 30 min | 27–30 | | 0.5–96 h | 20 min | 2 days | Twice: ~84% Once: ~60% | Associated with ↑MnSOD activity |
| Starnes | 2003 | Rat | M | Fischer 344 | 1 | 60 min | 14 | 70–75 | 24 h | 22.5 min | 30 min | Adults: ~80% Old: ~56% Young: ~39% | Not mediated by HSP70 or antioxidant enzymes |
| Domenech | 2002 | Dog | – | Mongrel | 1 | 25 min[5] | 100 | | 10 min vs. 24 h | 60 min | 270 min | Early: 78% Late: 46% | Early SEP is mediated by mitochondrial K$_{ATP}$ channel |
| Quindry | 2007 | Rat | M | Sprague-Dawley | 1 | 60 min | 30 | | 24 h | 50 min | 120 min | 67% | Not mediated by HSP72 |
| Yamashita | 2002 | Rat | M | Wistar | 1 | 20 min | 23–27 | | 24 h | 35 min | 120 min | 65% | Mediated by PKC activity |
| McGinnis | 2015 | Mouse | M | C57/Bl6 vs. IL-6$^{-/-}$ | 3 | 60 min | 18 | | 24 h | 30 min | 120 min | 64% | ↑STAT3, MAPK, IL-6, no change in eNOS, COX-2 |
| Lennon (b) | 2004 | Rat | M | Sprague-Dawley | 3 | 60 min | | 55 vs. 75 | 24 h | 20 min | 30 min | MIIT: ~62%[3] HIIT: ~56%[3] | Associated with ↑MnSOD and Hsp72 |
| Melling | 2009 | Rat | M | Sprague-Dawley | 1 | 60 min | 30 | | 5–10 min vs. 24 h | 30 min | 30 min | 60% | Mediated by PKC activity |
| Yamashita | 1999 | Rat | M | Wistar | 1 | 30 min | 30 | | 0.5–60 h | 20 min | 2 days | Until ~60% | Mediated by TNFα, IL-1β and Mn-SOD |
| Taylor | 2012 | Rat | M | Fischer 344 | 1 | 60 min | 20 | | 24 h | 22.5 min | 30 min | 56%[3] | Not mediated by ROS |
| Parra | 2010 | Dog | – | Mongrel | 1 | 25 min[5] | 100 | | 24 h | 60 min | 270 min | 50% | Late SEP is mediated by mitochondrial K$_{ATP}$ channel |
| Michelsen[4] | 2012 | Rabbit | M | New Zealand white | 1 | 20 min[5] | | 250–400 W | 5 min | 40 min | 120 min | 42% | Mediated by opioid receptor activation |

*(Continued)*

**Table 1. (Continued)**

| | General information | | | | Exercise protocol | | | | | Injury model | | Outcomes | |
| --- | --- | --- | --- | --- | --- | --- | --- | --- | --- | --- | --- | --- | --- |
| Reference | Year | Species | Sex[1] | Strain | Repeat | Period[2] | Speed (m/min)[3] | Intensity ($\%\dot{V}_{O_2max}$) | Time since exercise | Ischaemia | Reperfusion | Protection | Mechanism |
| Akita | 2007 | Mouse | M | C57/Bl6 | 7 | 60 min | | 60–70 | 24 h | 30 min | 120 min | ~38% | ↑Sympathetic nerve activation, ROS, eNOS, iNOS |
| Ramez | 2020 | Rat | M | Wistar | 5 | 22 min[5] | | 85–90 | 24 h | 30 min | 1 day | 34% | Associated with ↑ Klotho, anti-oxidant enzymes and TRPC6 |
| Lennon (a) | 2004 | Rat | M | Sprague-Dawley | 3 | 60 min | | 70 | 24 and 72 h | 20.5 min | 30 min | ~29%[3] | Not mediated by catalase and HSP72 activity |
| **Other types of IR-injury models** | | | | | | | | | | | | | |
| Hafez | 2020 | Rat | M | Wistar | 1 | 30 min[5] | 30 | | 1 h | Thrombo-embolic stroke model | | 74% | Mediated by eNOS activity, associated with AMPK activity |
| Sun | 2014 | Rat | M | Sprague-Dawley | 3 vs. 21 | 45 min[5] | 28–30 | 75 | 24 h | Isoproterenol-induced myocardial injury model | | Short: 55%[3] Long: 68%[3] | - |
| Reger | 2012 | Rat | F | Wistar-Kyoto vs. SHR[6] | 1 | 60 min | 25 | | 1 h | 22 min | None | Increased peak contracture during ischaemia for SHR rats | – |

[1] M = male, F = female.
[2] Warm-up period and acclimatization protocols are excluded, in minutes.
[3] % increase in cardiac work recovery instead of infarct size.
[4] Using human plasma dialysate from exercising healthy volunteers.
[5] Exercise performed with interval sessions.
[6] Spontaneously hypertensive rats.

(on three consecutive days) in 26 patients with rest angina who demonstrated coronary spasm (Morikawa *et al.* 2013). Remarkably, only three sessions of interval exercise significantly suppressed the number of coronary spasms, an effect that seems unlikely to be mediated through structural cardiovascular adaptations and/or individual risk factors.

Another observation in humans relates to the increase in cardiac troponin, i.e. a marker of myocardial injury, found after strenuous exercise. Recent work revealed that a single session of endurance exercise is protective against the exercise-induced rise in cardiac troponin T. In two other studies from separate laboratories, it was shown that cTnT was blunted during a second exercise session that was separated by both 4 and 48 h of recovery in previously inactive, young women (Nie *et al.* 2019; Zhang *et al.* 2019). These findings, although indirectly, provide some support for the concept that SEP leads to smaller tissue injury. Furthermore, Michelsen *et al.* (2012) investigated blood samples from healthy participants following a single bout of high-intensity interval cycling or IPC. Using rabbit Langendorff-perfused hearts, it was shown that both exercise and IPC reduced myocardial infarct size by ∼50%, further supporting the acute preconditioning effects of exercise.

A more frequently applied human *in vivo* model of IR injury relates to a surrogate model involving 15–20 min of ischaemia induced in the arm by inflation of a blood pressure cuff, followed by 15–20 min of reperfusion (deflation of cuff). This method induces impairments in brachial artery function as measured by flow-mediated dilatation (FMD) (Kharbanda *et al.* 2001), a measure with strong correlation to coronary artery responses (Anderson *et al.* 1995; Broxterman *et al.* 2019). Using this surrogate model, Seeger *et al.* (2015) showed that a single session of high-intensity interval exercise prior to ischaemia, but not continuous isocaloric exercise, attenuated the decline in FMD from whole-arm IR injury in young healthy volunteers. An important observation was that the efficacy of preconditioning seems attenuated in subjects with cardiovascular risk factors or disease (Ferdinandy *et al.* 2014; Seeger *et al.* 2016). Interestingly, in subsequent studies, it was demonstrated that habitually endurance-trained middle-aged to elderly individuals show smaller endothelial IR injury compared to their sedentary counterparts (Devan *et al.* 2011*a*; Maessen *et al.* 2017). More recently, our laboratory found exaggerated endothelial injury to IR and attenuated efficacy of preconditioning in patients with heart failure (Seeger *et al.* 2016). Twelve weeks of either continuous or HIIT mitigated the endothelial IR injury (Thijssen *et al.* 2019). Table 2 details studies performed in humans that assessed the impact of acute exercise and habitual exercise training on protection against vascular IR injury.

In conclusion, these studies in humans suggest that short-term exercise induces immediate protection against (surrogate models of) cardiac and vascular injury. The efficacy of SEP in ageing and diseased populations suggests that SEP may be a more robust stimulus as compared to IPC. Evidence is appearing that repeated bouts of continuous exercise, as demonstrated in heart failure patients, bolster the protective effects of SEP. Although larger and more heterogenous human studies of SEP are required, especially for understanding the time course of exercise-induced cardioprotection, single or short-term exercise may represent a powerful tool for cardioprotection.

## What are potential effect modulators of exercise-induced cardioprotection?

The strength of SEP may depend on factors related to the exercise stimulus itself or to subject-related factors (Fig. 2). Below, the most important factors are discussed.

### Cardiovascular risk and disease

In line with preclinical data in preconditioning, the presence of cardiovascular risk factors or disease can attenuate the efficacy of SEP. This is important because loss of efficacy of the preconditioning stimulus predominates the lack of translation to clinical implementation. For example, the benefits of acute exercise against ischaemic injury were absent in spontaneously hypertensive rats as opposed to healthy rat hearts in an isolated Langendorff-perfused heart model (Reger *et al.* 2012). Similarly, older humans and those with heart failure demonstrate diminished exaggerated endothelial IR injury, whilst responsiveness to classic ischaemic preconditioning is attenuated (van den Munckhof *et al.* 2013; Seeger *et al.* 2016).

Importantly, the trained heart appeared to restore age-mediated loss of IPC, possibly through increased noradrenaline release (Abete *et al.* 2000). To support the benefits of exercise, Starnes *et al.* (2003) reported that exercise-induced cardioprotection was similarly occurring in untrained young, adult and aged rats (Starnes *et al.* 2003). Also in humans, cross-sectional data support that regular exercise is associated with lower endothelial IR injury in older subjects. Importantly, 12 weeks of either interval or continuous exercise training appeared to provide protection against vascular injury (Thijssen *et al.* 2019). This suggests that regular exercise training attenuated endothelial IR injury and improved the efficacy of the SEP-related 'preconditioning' stimulus in those with cardiovascular risk factors or disease. Exploring whether shorter durations of exercise can offer similar protection is warranted. The ability to repeatedly expose individuals to the SEP stimulus, and the suggestion that

**Table 2. Effects of acute, short-term, and habitual exercise on cardiac and vascular protection against ischemia-reperfusion injury in humans**

| Reference | Population characteristics | Study design | Exercise intervention (intensity, duration, mode) | Results |
|---|---|---|---|---|
| **Acute exercise** | | | | |
| Michelsen et al. (2012) | 11 healthy male volunteers 18–40 years | Randomized, cross-over, control | High-intensity interval cycling: Four, 2 min bouts of 250–400 W, 3 min recovery periods | Plasma dialysate obtained after exercise bout reduced myocardial infarct size of isolated perfused rabbit hearts undergoing IR by ~50% |
| Seeger et al. (2015) | 17 healthy volunteers (7 women), 23 ± 4 years | Randomized, cross-over, control | High-intensity interval cycling: Ten, 1 min bouts, 100% maximum workload (determined by incremental exercise test), 2 min recovery periods Continuous isocaloric cycle exercise: 28 min, 50% maximum workload | No change in brachial artery FMD (%) following whole-arm IR injury when preceded with interval (7.7 ± 3.1 to 7.2 ± 3.1) but not continuous exercise (7.8 ± 3.1 to 3.8 ± 1.7) |
| **Short-term – habitual exercise training** | | | | |
| Thijssen et al. (2019) | 20 heart failure patients (1 woman) 64 ± 8 years | Randomized, control | 12 weeks High-intensity interval cycling: Ten, 1 min bouts, 90% maximum workload, 2.5 min recovery periods Continuous cycle exercise: 30 min, 60–75% maximum workload | Both interval and continuous exercise training mitigated IR injury-induced impairments in brachial artery FMD |
| Maessen et al. (2017) | Older men 18 sedentary, 63 ± 7 years 20 veteran athletes, 63 ± 6 years | Cross-sectional | Endurance-trained athletes: >5 h of exercise/week for more than 20 years Sedentary individuals: <1 h of exercise/week for more than 20 years | No change in brachial artery FMD following whole-arm IR in lifelong athletes [3.0 (1.7–5.4) to 3.0 (1.9–4.1)] In sedentary individuals, FMD was significantly reduced following IR [3.0 (2.0–4.7) to 2.1 (1.5–3.9)] |
| DeVan et al. (2011)a | Young (n = 19, 18–40 years) and middle-aged (n = 18, 41–65 years) men and women Healthy sedentary and habitually exercise-trained volunteers | Cross-sectional | Endurance-trained volunteers: habitually performed cycling and/or running at a moderate to strenuous exercise intensity for 8.6 ± 0.7 h/week verified by $\dot{V}_{O_2max}$ testing Sedentary volunteers: no exercise or <2 h/week for the past year | Decline in FMD following lower-arm IR injury was less in endurance-trained middle-aged individuals (50%) vs. sedentary age-matched counterparts (68%) |
| DeVan et al. (2011)b | Young men and women 11 sedentary (2 women), 26 ± 2 years 11 habitually trained (1 woman), 25 ± 2 years | Cross-sectional | Resistance-trained individuals: lifted weights targeting all major muscle groups >2 times/week for >1 year Sedentary volunteers: no exercise or <2 h/week for the past year | Significant decline (36%) in brachial artery FMD following lower arm-IR injury in sedentary participants but not in resistance-trained participants |

this improves the efficacy of the cardioprotective stimulus, offers important advantages.

### Sex

Whilst exercise has beneficial effects for men and women, isolated hearts subjected to regional ischaemia and reperfusion showed infarct size reduction 1 day after a single exercise bout in male but not female rats (Brown *et al.* 2005). Increasing the frequency to five bouts led to infarct sparing for both male and female hearts. Importantly, infarct size was already >25% lower in female *vs.* male sedentary hearts, suggesting a smaller window for cardioprotection in female hearts. These differences may relate to higher oestrogen levels in women.

### Exercise moderating factors

**Mode.** Studies evaluating SEP in both human and animal models have largely centred on endurance continuous training such as swimming and running. Few studies have addressed the effects of resistance training on mitigating myocardial injury. In a cross-sectional study, DeVan *et al.* (2011*b*) showed that young individuals who participated in habitual resistance training demonstrated improved vascular protection against endothelial IR injury in comparison to sedentary age-matched volunteers. Resistance training for 12 weeks in rats resulted in attenuated myocardial infarct size following IR injury and preserved cardiac function during IR in comparison to sedentary controls (Soufi *et al.* 2011) However, in another study 4 weeks of similar exercises (5 days/week) did not reveal such protection (Doustar *et al.* 2012).

Further investigation into the time-course of resistance training-induced cardioprotection is warranted as this may represent a type of training that can induce down-stream ischaemia, affording remote protection to the vasculature, and may be more easily incorporated into practice for some populations.

**Intensity.** Comparisons of HIIT and moderate-intensity continuous aerobic exercise largely derive from human work, suggesting that a single session of high-intensity interval exercise has more protective effects than continuous exercise in young individuals (Seeger *et al.* 2015). Although no direct comparison was made, Morikawa *et al.* (2013) found that 3 days of aerobic interval training successfully decreased the number of coronary spasms in patients with rest angina. However, in heart failure patients, the discrepancy between interval and continuous exercise disappears with a longer duration of training (12 weeks) (Thijssen *et al.* 2019). In rats, it appears that both moderate continuous and high-intensity exercise exert similar cardioprotective benefits against IR injury. Rats that exercised for 3 days (1 h/day) at either ~55% or ~75% of maximal oxygen uptake demonstrated equivalent preserved cardiac work and pre-ischaemic cardiac output during reperfusion, in comparison to sedentary rats (Lennon *et al.* 2004*b*).

**Duration.** A recent study examined whether exercise duration influences the degree of cardioprotection. Comparing responses to myocardial IR following brief (2 days), intermediate (7–14 days) and extended (28 days) voluntary wheel running, male mice showed improved tolerance to IR by 7 days, with no further improvements with increasing duration or volume of

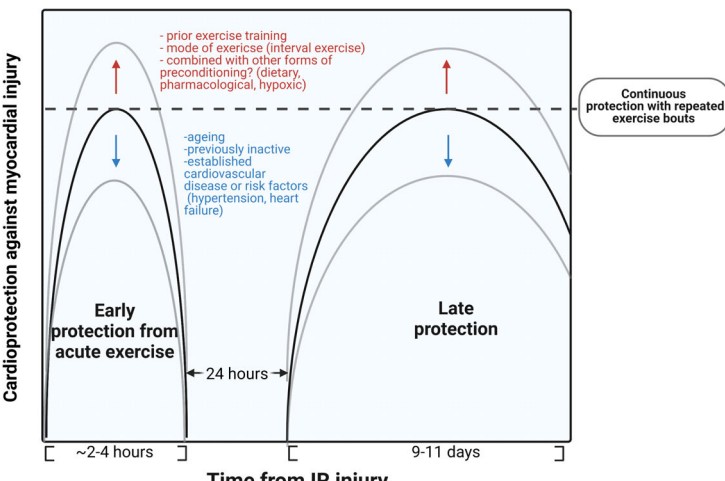

**Figure 2. Proposed modulators of short-term exercise-induced protection (SEP)**
Exercise stimulus (intensity, mode) and subject-related factors (ageing, prior training status, disease risk factors) may influence the magnitude of exercise-induced cardioprotection against myocardial injury.

exercise (Budiono *et al.* 2021). While running volume and intensity were not standardized in this study, these findings are in line with earlier work demonstrating comparable effects of 3 days and 3 weeks of intermittent running in protecting against isoproterenol-induced myocardial injury in rats (Sun & Pan, 2014). Whether these findings can be translated to populations with (risk factors for) cardiovascular disease warrants investigation

## Which effector signalling cascades are responsible for SEP?

Since the haemodynamic alterations of exercise employed in the acute setting are transient (Parra *et al.* 2015), these direct cardiac changes are unlikely to account for the infarct sparing effect of SEP. In an open-chest pig myocardial infarction model, previous induction of tachycardia through rapid pacing provided modest but significant cardioprotection (infarct size control 84% *vs.* pacing 79%; Koning *et al.* 1996). This suggest that increased heart rate during exercise per se cannot fully explain the preconditioning effects of SEP. Vascular alterations, such as increased collateral recruitment, have been observed after exercise in dogs as well as in cardiac patients who underwent percutaneous coronary intervention (PCI). However, the effects of SEP do not seem to rely on collateralization to generate cardioprotection (Lambiase *et al.* 2003; Parra *et al.* 2010, 2015). In discussing the potential underlying mechanisms, we have separated the effector signalling cascades (probably deriving from exercising muscles) and intracellular signalling pathways (in the effector organs; heart, vasculature; Fig. 3).

### Humoral factors

Circulating humoral factors, including adenosine, bradykinin and opioids, have been shown to be eminent in various preconditioning strategies (Heusch *et al.* 2015; Wang *et al.* 2021; Wu *et al.* 2021). These messenger molecules activate myocardial G-coupled receptors and activate downstream cell signalling leading to increased nitric oxide formation, protein kinase C activity and eventual opening of the ATP-sensitive $K^+$-channel (further described below). The contribution of humoral factors in exercise preconditioning has been sparsely investigated. In isolated Langendorff-perfused rabbit hearts perfused with human plasma dialysate, Michelsen *et al.* (2012) suggested that SEP may be a remote phenomenon, which is transferrable across two different species. To better understand and identify the potential blood-borne humoral factors, these authors repeated their experiments in the presence of naloxone, a selective $\mu$-opioid receptor antagonist. Interestingly, the protection of exercise disappeared when combined with

naloxone, suggesting that opioid receptor activation is eminent for cardiac SEP (Michelsen *et al.* 2012).

### Proinflammatory cytokines

The heart is instantly exposed to a sublethal amount of stressors by exercise, which are released at the site of one or more muscle groups or by the heart itself (Whitham *et al.* 2018; Contrepois *et al.* 2020). Increased myocardial levels of tumour necrosis factor $\alpha$ (TNF$\alpha$) and interleukin 1$\beta$ (IL-1$\beta$) immediately after exercise in rats and mice have been previously reported (Yamashita *et al.* 1999; McGinnis *et al.* 2015). The same authors later reported that TNF$\alpha$ administration confers cardioprotection in a biphasic manner, lasting up until 72 h after the IR insult (Yamashita *et al.* 1999; Hoshida *et al.* 2002). Whether cytokines released by muscle tissue is mediating SEP has not been directly investigated, but infusing TNF$\alpha$ systemically at 30 min and 48 h before IR caused robust infarct size lowering similar to exercise (Yamashita *et al.* 1999). It may therefore be plausible that TNF$\alpha$ is one of the signalling molecules conferring SEP during the first and second window of protection. Nevertheless, SEP was not attenuated with blockage of TNF$\alpha$ at the aforementioned time points, but only when combined inhibition of TNF$\alpha$ and IL-1$\beta$ occurred, suggesting that a single type of cytokine seems insufficient for exercise-induced cardioprotection in rats. Following 3 days of treadmill exercise, cardiac infarct size after IR was lowered in C57Bl/6 mice, an effect that was absent in IL-6$^{-/-}$ mice (McGinnis *et al.* 2015). This finding implies that also IL-6 is involved in SEP. Upregulation of serum IL-6 levels peaking at 30 min after exercise was observed in wild-type mice, as well as increased STAT3 phosphorylation (McGinnis *et al.* 2015). Myocardial STAT3 phosphorylation was, however, also detected in exercised IL6$^{-/-}$ mice. Interestingly, in IL6$^{-/-}$ animals undergoing IPC, STAT3 signalling and protection by IPC were missing, suggesting a divergence in IL6/STAT3 signalling between SEP and IPC (Dawn *et al.* 2004). In conclusion, pro-inflammatory cytokines, including TNF$\alpha$, IL-1$\beta$ and IL-6, are dynamically involved in SEP. Yet, whether these cytokines originate from exercise muscles rather than from the myocardium, and the inter-linkage of these cytokines, remain unresolved.

### Circulating metabolites

Changes in plasma metabolites may play a prominent role in acute SEP by modulating cardiac metabolism (Gibb *et al.* 2017; Contrepois *et al.* 2020; Zuurbier *et al.* 2020). In a metabolic profiling study with Framingham Heart participants, over 80% of the total 588 studied metabolites were altered after a single cardiopulmonary exercise test (Nayor *et al.* 2020). Increased plasma lactate levels

(Michelsen *et al.* 2012) and increased insulin sensitivity (Newsom *et al.* 2013; Short *et al.* 2018) are some of the acute changes following exercise in obese and healthy human subjects. In mice performing SEP, lower plasma glucose levels and acutely increased plasma free fatty acids after the exercise bout have been observed (Gibb *et al.* 2017). However, evidence on the causality of these plasma-derived exercise-induced metabolic alterations towards cardioprotection remains rather limited.

## Myokines

A more recently identified group of mediators are muscle-derived signalling proteins, which could have anti-apoptotic and anti-inflammatory effects that favour attenuated IR injury. Myonectin, a myokine belonging to the C1q/TNF-related protein (CTRP) family, was recently reported to lower cardiac IR injury in response to 4 weeks of exercise training (Otaka *et al.* 2018). Accordingly, it would be worthwhile to investigate whether myonectin is involved in SEP. Another myokine, MG53, was reported to be upregulated by IPC in a PKCδ-dependent manner (Shan *et al.* 2020). MG53 has been previously associated with exercise performance (Cai *et al.* 2009), although the link between SEP and MG53 as a circulating myokine is as yet unexplored. These findings may put myokines on the map for future investigations in their role during SEP.

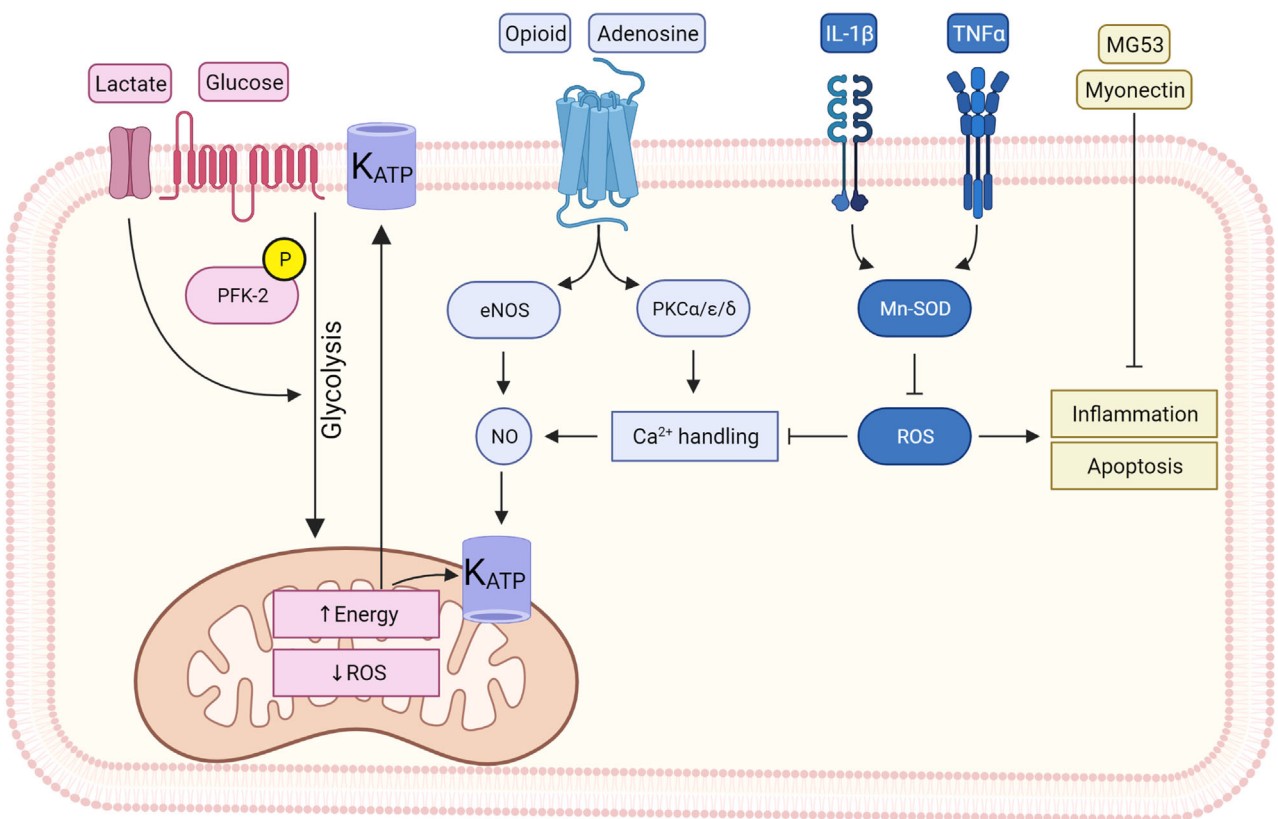

**Figure 3. Proposed effector signalling and intracellular mechanisms of SEP**
Release of myokines, metabolites, humoral factors, cytokines and ROS in the circulatory system by exercise modulates intramyocardial signalling during IR, contributing to reduced inflammation and apoptosis, increased anti-scavenging processes and $K_{ATP}$ channel activation, which ultimately result in protection against cardiac cell death.

# What intracellular signalling pathways protect against cell death in SEP?

## Anti-oxidative signalling

The anti-oxidative signalling after SEP can be described as a process of hormesis: the concept that substances are detrimental at higher levels, but have protective effects at low dosage. Cardiac proteome analyses in healthy and obese mice showed reduced levels of the mitochondrial enzyme aconitase following exercise, indicating increased reactive oxygen species (ROS) formation (Petriz *et al.* 2013). This may imply that a single bout of exercise upregulates levels of ROS that could evolve to activation of anti-oxidative signalling. Indeed, in wildtype mice, acute exercise exerted ROS formation prior to activation of anti-oxidative defence pathways, an effect that was modulated by Nrf2, a basic leucine zipper protein responsible for anti-oxidative gene expression (Muthusamy *et al.* 2012). Increased activity of manganese superoxide dismutase (Mn-SOD), an intrinsic scavenger of superoxide anions, was observed following one or two exercise sessions and seemed to be associated with cardioprotection in mice and rats (Yamashita *et al.* 1999; Hoshida *et al.* 2002; Hamilton *et al.* 2004; Muthusamy *et al.* 2012). Inhibiting Mn-SOD abolished exercise-induced cardioprotection. Furthermore, Mn-SOD appeared to be activated by either Il-1$\beta$ or TNF$\alpha$, while simultaneous inhibition of these cytokines inhibited Mn-SOD activity and SEP against myocardial IR injury in rats (Yamashita *et al.* 1999). These data suggest that the immediate upregulation of Il-1$\beta$ and/or TNF$\alpha$ by exercise enhances Mn-SOD activity during SEP. Additionally, a single HIIT session preserved the anti-oxidant system during IR and upregulated the anti-ageing protein Klotho in male Wistar rats (Ramez *et al.* 2020). Using a different exercise protocol and a shorter reperfusion period, contrasting results were observed by Taylor & Starnes (2012), reporting that SEP-induced changes to cardiac mechanical function and lactate dehydrogenase (LDH) release were not affected by administering the free radical scavenger N-(2-mercaptopropionyl)glycine (MPG) in male Fischer 344 rats. Others reported that infarct size lowering by SEP was absent when preceded with MPG administration in male Wistar rats (Yamashita *et al.* 1999). Taken together, a distinct pattern of anti-scavenging signalling is observed in SEP, which appears following pro-inflammatory cytokines and ROS production.

## K$_{ATP}$ channels

Regulated by intracellular nucleotide, ADP and ATP levels, the K$_{ATP}$ channels serve as energy sensors of the cell. Activation of the cardiac sarcolemmal or mitochondrial K$_{ATP}$ channels is a common denominator in mice and dogs that performed SEP (Domenech *et al.* 2002; Quindry *et al.* 2010). Increased K$_{ATP}$ channel expression by exercise resulted in enhanced action potential shortening and improved cardiac energy consumption under increasing cardiac workload in mouse ventricles (Zingman *et al.* 2011). These effects were eliminated in transgenic mice with cardiac-specific disruption of K$_{ATP}$ channel function, suggesting that K$_{ATP}$ channel function mediates the cardiac energy-saving response of exercise. Interestingly, involvement of sarcolemmal, but not mitochondrial, K$_{ATP}$ channels in SEP was confirmed after prolonged IR injury in rats. Further studies are needed to understand the relevance of K$_{ATP}$ channel activation to improve outcomes of myocardial IR injury by SEP.

## Nitric oxide synthase systems

Nitric oxide synthases (NOS) are a set of enzymes that convert arginine to nitric oxide, which is a multi-faceted cellular cross-talk molecule related to vasodilatory response, thrombogenicity, proliferation, inflammation, contractility and mitochondrial function. The significance of NOS for SEP was demonstrated in C57BL/6 mice undergoing 7 days of exercise (Akita *et al.* 2007). Increased activity of endothelial NOS (eNOS) and expression of inducible NOS (iNOS) were observed in exercised animals. While exercise effectively attenuated infarct size following myocardial IR in wild-type mice, eNOS$^{-/-}$ and iNOS$^{-/-}$ variants were resistant to the exercise-induced cardioprotection (Akita *et al.* 2007). In humans, 3 days of interval exercise may largely improve NO-mediated, flow-mediated dilatation in patients with rest angina (Morikawa *et al.* 2013), suggesting a potential role for NOS systems (and its relation to endothelial function) in SEP.

## PKC channels

Protein kinase C (PKC) acts as a signalling molecule for cellular ion homeostasis, calcium handling, contractile function and cardioprotection. In the early window following one and seven bouts of exercise, it appears that PKC isoforms are bi-directionally modulated in rat hearts (Carson & Korzick, 2003). PKC$\alpha$, known for its role in myocardial hypertrophy, is enhanced immediately after one exercise bout, while seven consecutive bouts lead to its reduction. On the other hand, PKC$\varepsilon$ levels were reported to be elevated after one and seven bouts of exercise. Upregulation of PKC$\delta$ was observed immediately and 24 h after exercise (Melling *et al.* 2009). Interestingly, a PKC$\alpha$/$\varepsilon$/$\delta$ inhibitor blunted exercise-induced cardioprotection in isolated IR injury hearts (Melling *et al.* 2009), suggesting that at least one PKC isoform may be regulating

SEP. Further investigations towards the relevance of the different PKC isoforms on SEP are needed.

## Heat shock proteins

Heat shock proteins (HSPs) cover a large family of molecular chaperones that are triggered by exposure to stress. Studies have reported increased HSP70/72 expression by exercise as a potential source for cardio-protection following training (Locke *et al.* 1995; Quindry *et al.* 2007). In contrast, HSP70 induction by exercise was only modestly present in adult and senescent IR hearts (Starnes *et al.* 2003). These findings have proposed ageing as a modifier of HSP70 expression. Yet, because cardio-protection is observed in all age categories, HSP70 may play a minor role as a mediator of exercise-induced cardio-protection. Others have reported that HSP upregulation acts merely as an epiphenomenon during SEP (Hamilton *et al.* 2001; Quindry *et al.* 2007). Taken together, data imply that there is an association between HSPs and single exercise, but that HSPs may not mediate the protective effects of SEP.

## Conclusion and future perspective

Evidence that regular exercise is beneficial for lowering cardiovascular-related morbidity and mortality is well established, yet how fast we achieve these benefits is less clear. Data presented in this review suggest that some of the benefits of exercise, especially related to the magnitude of damage induced by IR injury, are present after a single or short-term period. Relatively little work has been performed in this new area. Hence, important questions relate to the temporal aspect of SEP, i.e. when and how long is SEP present. While animal studies serve a clear purpose in examining SEP, translation of these findings to human populations can be poor. Current knowledge is largely based on small-sized studies, typically evaluating end-urance exercise in rats. Use of large animals, specifically pigs, would be well suited to bridge this translational gap (Lieder *et al.* 2018; Skyschally *et al.* 2019). Furthermore, modifying factors on the degree of cardioprotection, including exercise type or different patient populations, can be evaluated in humans when combined with a bio-assay to assess cardioprotection using isolated hearts or cardiomyocytes (Michelsen *et al.* 2012; Skyschally *et al.* 2015, 2018; Lieder *et al.* 2019). Finally, SEP represents an attractive intervention in aged individuals and in those with co-morbidities. The immediate protection, low cost and simplicity to increase the 'dose' of SEP offers unique opportunities in the clinical application of SEP. Capitalizing on better understanding the mechanisms and key questions raised above, studies should explore the feasibility and efficacy of SEP-based interventions to improve clinical outcome and quality of life of patients with cardiovascular disease.

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

## Additional Information

### Competing interests

None.

### Author contributions

All authors approved the final version of the manuscript and qualify for authorship, and all those who qualify for authorship are listed. All authors conceptualized the outline of the review. The manuscript and figures were prepared by L.U. and Y.S., with help from D.T. and N.v.R. All authors edited and commented on the text and figures.

### Funding

Y.S. is funded through the Canadian Institutes for Health Research (CIHR).

### Keywords

cardioprotection, ischaemia reperfusion injury, ischaemic heart disease, single exercise-induced protection

### Supporting information

Additional supporting information can be found online in the Supporting Information section at the end of the HTML view of the article. Supporting information files available:

**Peer Review History**

