## [Peer Review History · The Journal of Physiology]

Understand and underestimating exercise as medicine: Why and how fast does the heart benefit from exercise?

Dick H.J. Thijssen, Laween Uthman, Yasina Begum Somani, and Niels van Royen

DOI: 10.1113/JP282000

Corresponding author(s): Dick Thijssen (dick.thijssen@radboudumc.nl)

The following individual(s) involved in review of this submission have agreed to reveal their identity: Shane A Phillips (Referee #1); Huub - Maas (Referee #2)

Review Timeline:

Submission Date:	18-Jun-2021
Editorial Decision:	09-Aug-2021
Revision Received:	01-Nov-2021
Accepted:	10-Dec-2021

Senior Editor: Ian Forsythe

Reviewing Editor: Simon Gandevia

Transaction Report:

Dear Dr Thijssen,

Re: JP-SR-2021-282000 "Understand and underestimating exercise as medicine: Why and how fast does the heart benefit from exercise?" by Dick H.J. Thijssen, Laween Uthman, Yasina Begum Somani, and Niels van Royen

Thank you for submitting your invited Symposium Review to The Journal of Physiology. It has been assessed by a Reviewing Editor and by 2 expert referees and I am pleased to tell you that it is considered to be acceptable for publication following satisfactory revision.

The reports are copied at the end of this email. Please address all of the points and incorporate all requested revisions, or explain in your Response to Referees why a change has not been made.

NEW POLICY: In order to improve the transparency of its peer review process The Journal of Physiology publishes online as supporting information the peer review history of all articles accepted for publication. Readers will have access to decision letters, including all Editors' comments and referee reports, for each version of the manuscript and any author responses to peer review comments. Referees can decide whether or not they wish to be named on the peer review history document.

I hope you will find the comments helpful and have no difficulty in revising your manuscript within 4 weeks.

Your revised manuscript should be submitted online using the links in Author Tasks Link Not Available. This link is to the Corresponding Author's own account, if this will cause any problems when submitting the revised version please contact us.

The image files from the previous version are retained on the system. Please ensure you replace or remove any files that have been revised. Your revised submission should include:

- A Word file of the complete text (including figure legends any Tables);
- An Abstract Figure (with legend in the Article file)
- Each figure as a separate, high quality, file;
- A full Response to Referees;
- A copy of the manuscript with the changes highlighted.
- Author profile. A short biography (no more than 100 words for one author or 150 words in total for two authors) and a portrait photograph of the two leading authors on the paper. These should be uploaded, clearly labelled, with the manuscript submission. Any standard image format for the photograph is acceptable, but the resolution should be at least 300 dpi and preferably more.

- A 'Cover Art' file for consideration as the Issue's cover image;
- Appropriate Supporting Information (Video, audio or data set https://jp.msubmit.net/cgi-bin/main.plex?form_type=display_requirements#supp).

To create your 'Response to Referees' copy all the reports, including any comments from the Reviewing Editor into a Word, or similar, file and respond to each point in colour or CAPITALS and upload this when you submit your revision.

I look forward to receiving your revised submission.

If you have any queries please reply to this email and staff will be happy to assist.

Yours sincerely,

Ian D. Forsythe
Deputy Editor-in-Chief
The Journal of Physiology
<https://jp.msubmit.net>
<http://jp.physoc.org>
The Physiological Society
Hodgkin Huxley House
30 Farringdon Lane
London, EC1R 3AW
UK
<http://www.physoc.org>
<http://journals.physoc.org>

REQUIRED ITEMS:

-Your MS must include a complete "Additional information section" with the following 4 headings and content:

Competing Interests: A statement regarding competing interests. If there are no competing interests, a statement to this effect must be included. All authors should disclose any conflict of interest in accordance with journal policy.

Author contributions: Each author should take responsibility for a particular section of the study and have contributed to writing the paper. Acquisition of funding, administrative support or the collection of data alone does not justify authorship; these contributions to the study should be listed in the Acknowledgements. Additional information such as 'X and Y have contributed equally to this work' may be added as a footnote on the title page.

It must be stated that all authors approved the final version of the manuscript and that all persons designated as authors qualify for authorship, and all those who qualify for authorship are listed.

Funding: Authors must indicate all sources of funding, including grant numbers. If authors have not received funding, this must be stated.

It is the responsibility of authors funded by RCUK to adhere to their policy regarding funding sources and underlying research material. The policy requires funding information to be included within the acknowledgement section of a paper. Guidance on how to acknowledge funding information is provided by the Research Information Network. The policy also requires all research papers, if applicable, to include a statement on how any underlying research materials, such as data, samples or models, can be accessed. However, the policy does not require that the data must be made open. If there are considered to be good or compelling reasons to protect access to the data, for example commercial confidentiality or legitimate sensitivities around data derived from potentially identifiable human participants, these should be included in the statement.

Acknowledgements: Acknowledgements should be the minimum consistent with courtesy. The wording of acknowledgements of scientific assistance or advice must have been seen and approved by the persons concerned. This section should not include details of funding.

-Please upload separate high quality figure files via the submission form.

-Author profile(s) must be uploaded via the submission form. Authors should submit a short biography (no more than 100 words for one author or 150 words in total for two authors) and a portrait photograph of the two leading authors on the paper. These should be uploaded, clearly labelled, with the manuscript submission. Any standard image format for the photograph is acceptable, but the resolution should be at least 300 dpi and preferably more. A group photograph of all authors is also acceptable, providing the biography for the whole group does not exceed 150 words.

EDITOR COMMENTS

Reviewing Editor:

This is a timely review of a neat evolving concept. The submission is well written and easy to follow. The reviewers identify areas where there could be definite improvement to increase the real information content and ultimately the impact of the review. The figures are clear and helpful. However, I strongly recommend that the species be mentioned for particular research findings.

REFEREE COMMENTS

Referee #1:

This is an interesting review paper which suggests that there is a window of opportunity for exercise to improve and protect against cardiovascular risk and ischemia reperfusion injury. The overarching idea is that short term exercise protection attenuates the cardiac and vascular injury risk. The idea is novel and is supported by the literature reviewed in the paper. The primary concern is that the literature reviewed on intensity of SEP and CV protection lacks details on the specific exercise prescription. This makes it difficult to discern what the intensity level or duration of protections based on intensity might be. In addition, there is minimal discussion and review of mode of SEP. A table may help to guide the reader. Much of the discussion of the mechanisms of signaling is speculative and could be toned down. Lastly, the title does not seem clear. I would make it specific to short term exercise induced protection of CV function/health.

Referee #2:

The manuscript presents a review on a relatively new concept, i.e. Short-term Exercise induced Protection, which appears to result in cardiovascular protection. In three sections, the authors describe the evidence of SEP, the modulators of the protection and the potential underlying mechanism. The text is well structured and the literature is reviewed in a concise manner. Below my comments.

- The authors describe results from pre-clinical studies in which a variety of animal models are exploited. Reporting of the species, strain and sex is however not consistently done. At the minimum the species needs to be mentioned when describing these studies. The authors make only one remark about the results being transferrable across different species, but it is not clear if this is the case for all animal models described. Some critical notes regarding the potential for translation to humans seems warranted.

- The second key point is not a conclusion of this review. This could be replaced by the sentence in the abstract "SEP activates multiple pathways to confer...".

- Instead of ending the abstract with stating that future directions are discussed, it is more informative if the actual directions are described.

- In the section on mechanisms, several abbreviations are used but they are not consistently defined. Also a brief description of their general function like done for PKC would be helpful for the non-expert reader.

END OF COMMENTS

Confidential Review

18-Jun-2021

Dear Professor Kim E. Barrett, editor-in-chief of *the Journal of Physiology*,

We would like to submit the revision of the original manuscript entitled: entitled: *“Understanding and underestimating exercise as medicine: Why and how fast does the heart benefit from exercise?”*, now entitled *“Short term exercise-induced protection of cardiovascular function and health: Why and how fast does the heart benefit from exercise?”* as a symposium review accompanying the Second International Motor Impairment Conference.

The main feedback presented by the reviewers was related to reporting of relevant information from the studies presented in this manuscript. This includes the species, strain, age and exercise prescription. In the revised manuscript, we have included two tables that describe each of these factors, together with the ischemia-reperfusion injury protocol, the outcome of the study and potential mechanisms related to the observed protection. In addition, changes in the main text have been made accordingly, which better define the species type used in the studies that were discussed. We endeavor that these textual changes and the addition of the two tables will elaborate more on the current evidence for single-/short-term exercise-induced preconditioning (SEP) against cardiac ischemia/reperfusion injury.

Second, the manuscript has touched upon some of the mechanisms of SEP that were previously investigated. Since this area of preconditioning by exercise has been largely overlooked, as opposed to ischemic preconditioning strategies, most of the data exploring the mechanisms are retrieved from animal-based work. This makes it difficult to pinpoint intracellular pathways and effector signaling molecules related to the effectiveness of SEP. We have aimed to connect and describe some of the explored mechanisms in the original manuscript, but to ensure appropriate extrapolation and translation of these animal-based studies, we have refined the conclusions made on the mechanisms of SEP in this revised submission. In addition, we have included several suggestions for future studies that would benefit the translation of SEP.

This manuscript has not previously been published and is not under consideration elsewhere. I hereby confirm that all authors have approved this manuscript.

On behalf of all contributing authors,

Yours sincerely,

Dick Thijssen, (corresponding author)
Laween Uthman,
Yasina Somani
Niels van Royen

EDITOR COMMENTS

Reviewing Editor:

This is a timely review of a neat evolving concept. The submission is well written and easy to follow. The reviewers identify areas where there could be definite improvement to increase the real information content and ultimately the impact of the review. The figures are clear and helpful. However, I strongly recommend that the species be mentioned for particular research findings.

We thank the reviewing editor for the thorough review of our work. We have now included information regarding the species used in the mentioned studies. In addition, we have highlighted the species for each study in **table 1**, followed by the methods and outcomes and mechanisms explored by each study.

REFEREE COMMENTS

Referee #1:

This is an interesting review paper which suggests that there is a window of opportunity for exercise to improve and protect against cardiovascular risk and ischemia reperfusion injury. The overarching idea is that short term exercise protection attenuates the cardiac and vascular injury risk. The idea is novel and is supported by the literature reviewed in the paper. The primary concern is that the literature reviewed on intensity of SEP and CV protection lacks details on the specific exercise prescription. This makes it difficult to discern what the intensity level or duration of protections based on intensity might be. In addition, there is minimal discussion and review of mode of SEP. A table may help to guide the reader. Much of the discussion of the mechanisms of signaling is speculative and could be toned down. Lastly, the title does not seem clear. I would make it specific to short term exercise induced protection of CV function/health.

We thank the reviewer for the remarks and critical revision of our work. We agree with the lack of data regarding the exercise protocol implemented in the studies reviewed. Since there is minimal evidence on the mode of SEP in relation to ischemia/reperfusion injury, it is difficult to review and discuss that factor. However, we have now included two tables for the evidence from animal and human studies, respectively, where intensity, duration and the number of repetitions have been elaborated (**table 1 and 2**). In addition, we have made textual changes by adding a paragraph to discuss the duration of SEP (**pages 10-11**), the mechanisms, and have toned down speculative conclusions (throughout **pages 11-15**). We have also changed the title as suggested by the reviewer to: "*Short term exercise-induced protection of cardiovascular function and health: Why and how fast does the heart benefit from exercise?*".

Referee #2:

The manuscript presents a review on a relatively new concept, i.e. Short-term Exercise induced Protection, which appears to result in cardiovascular protection. In three sections, the authors describe the evidence of SEP, the modulators of the protection and the potential underlying mechanism. The text is well structured and the literature is reviewed in a concise manner. Below my comments.

We value the reviewer's critical revision of our manuscript. Please find our responses below each comment.

- The authors describe results from pre-clinical studies in which are variety of animal models are exploited. Reporting of the species, strain and sex is however not consistently done. At the minimum the species needs to be mentioned when describing these studies. The authors make only one remark about the results being transferrable across different species, but it is not clear if this is the case for all animals models described. Some critical notes regarding the potential for translation to humans seems warranted.

We agree with the reviewer that the information regarding the animal population used in the studies was inadequately reported in our previous manuscript. Therefore, we have included two tables summarizing the most important aspects for each human and animal study that was included and discussed in our manuscript (**table 1 and 2**). These tables includes information related to species, model and sex, but also exercise type, the outcome of the study in relation to cardioprotective effects, and the potential mechanisms that were examined in relation to SEP.

In only one study, the transfer of circulating effector signaling following SEP was evaluated in isolated perfused rabbit hearts [1]. To clarify that the transfer of SEP has so far only been demonstrated between two different species (rabbit and human), we have made textual arrangements stating that this transfer of SEP is transferable across two species (**page 11**).

Finally, we have highlighted the factors and potential perspectives for translation of animal-derived evidence of SEP by proposing the use of large animal models. Specifically, mechanistic explorations and translation to human populations seems feasible, whilst cross-species models (human-to-animal) are available to investigate feasibility of exercise in human subjects, while investigating the effectiveness of interventions against cardiac I/R-injury outcome in animal tissue (**page 16**). This information may be informative for those readers interested in the field and perform research in this area.

- The second key point is not a conclusion of this review. This could be replaced by the sentence in the abstract "SEP activates multiple pathways to confer...".

The second key point has now been adjusted to: *"SEP activates multiple pathways to confer cardiac protection, which develops remotely at the site of the activated muscle by release of circulating molecules, which transfer towards activation of intramyocardial signaling that promotes cell survival during episodes of IR injury."* (page 2).

We agree with the reviewer, as this statement should better fit as a conclusion derived from this manuscript.

- Instead of ending the abstract with stating that future directions are discussed, it is more informative if the actual directions are described.

We agree with this reviewer and we have changed the ending of the abstract to: *"Finally, we discuss potential future directions for designing adequate human and animal studies that will*

support developing effective SEP strategies for the (multi-)diseased and aged individual.”
(page 3).

- In the section on mechanisms, several abbreviations are used but they are not consistently defined. Also a brief description of their general function like done for PKC would be helpful for the non-expert reader.

We thank the reviewer for this observation. In the revised manuscript, the abbreviations are explained, and for each of the intracellular molecules an additional statement is added explaining the function (**pages 14-15**).

Reference

1. Michelsen, M.M., et al., *Exercise-induced cardioprotection is mediated by a bloodborne, transferable factor*. Basic Res Cardiol, 2012. **107**(3): p. 260.

Dear Professor Thijssen,

Re: JP-SR-2021-282000R1 "Understand and underestimating exercise as medicine: Why and how fast does the heart benefit from exercise?" by Dick H.J. Thijssen

Laween Uthman

Yasina Begum Somani

Niels van Royen

I am pleased to tell you that your Symposium Review article has been accepted for publication in The Journal of Physiology, subject to any modifications to the text that may be required by the Journal Office to conform to House rules.

NEW POLICY: In order to improve the transparency of its peer review process The Journal of Physiology publishes online as supporting information the peer review history of all articles accepted for publication. Readers will have access to decision letters, including all Editors' comments and referee reports, for each version of the manuscript and any author responses to peer review comments. Referees can decide whether or not they wish to be named on the peer review history document.

The last Word version of the paper submitted will be used by the Production Editors to prepare your proof. When this is ready you will receive an email containing a link to Wiley's Online Proofing System. The proof should be checked and corrected as quickly as possible.

All queries at proof stage should be sent to tjp@wiley.com

The accepted version of the manuscript is the version that will be published online until the copy edited and typeset version is available. Authors should note that it is too late at this point to offer corrections prior to proofing. Major corrections at proof stage, such as changes to figures, will be referred to the Reviewing Editor for approval before they can be incorporated. Only minor changes, such as to style and consistency, should be made a proof stage. Changes that need to be made after proof stage will usually require a formal correction notice.

Are you on Twitter? Once your paper is online, why not share your achievement with your followers. Please tag The Journal (@jphysiol) in any tweets and we will share your accepted paper with our 22,000+ followers!

Yours sincerely,

Ian D. Forsythe
Deputy Editor-in-Chief
The Journal of Physiology
<https://jp.msubmit.net>
<http://jp.physoc.org>
The Physiological Society
Hodgkin Huxley House
30 Farringdon Lane
London, EC1R 3AW
UK
<http://www.physoc.org>
<http://journals.physoc.org>

Comments:

Reviewing Editor:

The manuscript has been thoughtfully revised and hopefully will be a useful resource for researchers in the future.

REFeree COMMENTS:

Referee #1:

This is a nice review of the impact of exercise on vascular function. I have no further comments.

Referee #2:

All comments were addressed adequately.

*** IMPORTANT NOTICE ABOUT OPEN ACCESS ***

Information about Open Access policies can be found here <https://physoc.onlinelibrary.wiley.com/hub/access-policies>

To assist authors whose funding agencies mandate public access to published research findings sooner than 12 months after publication The Journal of Physiology allows authors to pay an open access (OA) fee to have their papers made freely available immediately on publication.

You will receive an email from Wiley with details on how to register or log-in to Wiley Authors Services where you will be able to place an OnlineOpen order.

You can check if your funder or institution has a Wiley Open Access Account here <https://authorservices.wiley.com/author-resources/Journal-Authors/licensing-and-open-access/open-access/author-compliance-tool.html>

Your article will be made Open Access upon publication, or as soon as payment is received.

If you wish to put your paper on an OA website such as PMC or UKPMC or your institutional repository within 12 months of publication you must pay the open access fee, which covers the cost of publication.

OnlineOpen articles are deposited in PubMed Central (PMC) and PMC mirror sites. Authors of OnlineOpen articles are permitted to post the final, published PDF of their article on a website, institutional repository, or other free public server, immediately on publication.

Note to NIH-funded authors: The Journal of Physiology is published on PMC 12 months after publication, NIH-funded authors DO NOT NEED to pay to publish and DO NOT NEED to post their accepted papers on PMC.

1st Confidential Review

01-Nov-2021